# Learning Time-Aware Assistance Functions for Numerical Fluid Solvers

## Abstract

Improving the accuracy of numerical methods remains a central challenge in many disciplines and is especially important for nonlinear simulation problems. A representative example of such problems is fluid flow, which has been thoroughly studied to arrive at efficient simulations of complex flow phenomena. This paper presents a data-driven approach that learns to improve the accuracy of numerical solvers. The proposed method utilizes an advanced numerical scheme with a fine simulation resolution to acquire reference data. We, then, employ a neural network that infers a correction to move a coarse thus quickly obtainable result closer to the reference data. We provide insights into the targeted learning problem with different learning approaches: fully supervised learning methods with a naive and an optimized data acquisition as well as an unsupervised learning method with a differentiable Navier-Stokes solver. While our approach is very general and applicable to arbitrary partial differential equation models, we specifically highlight gains in accuracy for fluid flow simulations.

## 1 Introduction

Numerical methods are a central component of many disciplines and widely used for solving a variety of linear and nonlinear problems. One of the long-standing targets is fluid flow, which is renowned for its great diversity and complexity in terms of dynamics. Studies in computational fluid dynamics have focused on numerical simulations for such problems and invested huge efforts in solving spatio-temporal partial differential equations (PDEs) such as the Navier-Stokes equations, which represent the well-established physical model for fluids.

Traditional methods typically improve accuracy with fine discretizations both in space and time. While the methods and computing power for numerical simulation have seen advances in recent years, there is still a pressing need for better efficiency and accuracy. For most practical applications of computer simulations, we are still far away from fully resolving all necessary scales of nature around us (Verma et al., 2018; Cummins et al., 2018).

To tackle this problem, we propose a data-driven approach that "assists" a given numerical method to improve its accuracy. To this end, we introduce a first learning-based approach that puts special emphasis on the time dimension. We demonstrate two variants to achieve this goal in the context of fluids: a supervised version with an optimization algorithm for acquisition of temporally constrained correction data and an unsupervised version with a *differentiable PDE solver* that allows us to autonomously takes into account temporal information when training. We compare advantages and disadvantages of both approaches, and our experiments show that, using our trained models, the simulation accuracy of the given solver can be significantly improved. In all cases, our trained models yield improved dynamics, and the learned assistance function lets a coarse simulation reproduce the behavior of the reference data more closely. In particular, we demonstrate the improvements of our approach over ad-hoc learning approaches.

## 2 Related work

Data-driven approaches were shown to be highly effective for inferring unknown PDEs and coefficients of corresponding terms (Brunton et al., 2016). Their capabilities were investigated for different problems, among others the Navier-Stokes equations (Rudy et al., 2017; Schaeffer, 2017). Deep

learning methods were successfully applied for learning stencils of advection diffusion problems (Bar-Sinai et al., 2018), to discover PDE formulations (Long et al., 2017), and to analyze families of Poisson equations (Magill et al., 2018). While identifying governing equations represents an interesting and challenging task, we instead focus on a general method to improve the solutions of chosen solution spaces.

Due to the complexity and nonlinear nature of fluid dynamics, they remain a challenging problem, and accordingly data-driven approaches including deep learning have received a lot of attention to reach an efficient solution for such problems (Kutz, 2017). Particularly, a turbulence modeling has been a representative objective in this context (Duraisamy et al., 2015; Maulik & San, 2017; Beck et al., 2018; Mohan et al., 2019). Moreover, both steady (Guo et al., 2016) and unsteady (Morton et al., 2018) flows as well as multiphase (Gibou et al., 2018) flows have been investigated with deep learning based approaches. Additionally, the convolutional neural network (CNN) based methods were studied for airfoil flow problems (Thuerey et al., 2018; Zhang et al., 2018).

For fluid simulation, particularly in the field of computer graphics, data-driven approaches have been considered as an efficient alternative to replace computationally expensive steps of numerical processes (Ladický et al., 2015; Tompson et al., 2017). Moreover, deep learning, particularly with generative adversarial models (Goodfellow, 2016), has been used for efficiently synthesizing details within coarsely resolved simulations (Chu & Thuerey, 2017; Xie et al., 2018). In addition to scalar transport and smoke flows, liquid droplets have been efficiently tackled by learning stochastic models (Um et al., 2018). Due to their capabilities for transforming space-time data into reduced latent spaces, fluid simulations were processed and re-simulated with neural network (NN) models (Kim et al., 2019; Prantl et al., 2019). Motivated by the success of previous studies regarding data-driven approaches in numerical simulations, we aim for enabling the learning of general correction functions. Our aim differs from methods that synthesize details for low-resolution input data. As such, our method is orthogonal to super-resolution methods and shares similarities with methods to identify discontinuities in finite difference solutions (Ray & Hesthaven, 2018). However, we investigate the problem for challenging two-dimensional flow problems and propose a new approach for learning the correction bby taking into account temporal dynamics.

## 3 CORRECTING NUMERICAL SIMULATIONS

Numerical methods yield approximations of a smooth function $f(\mathbf{x})$ in a discrete setting with an approximation error of the form $O(h^k)$ where $h$ is the step size of the discretization. Naturally, higher order methods with larger $k$ are preferable but difficult to arrive at in practice. In our setting, the motions of fluids can be represented by a series of vector fields $\mathbf{v}(\mathbf{x})$ where bold vectors will denote two- or three-dimensional vector-valued functions in the following. The accuracy of the fluid motion likewise depends on the discretization error for $\mathbf{v}$. In the following, we will focus on Cartesian Eulerian representations, meaning axis aligned grids with square cells. Discretizing functions like $\mathbf{v}$ on such a grid yields approximations that scale with the grid spacing $h$. While small $h$ can yield accurate representations, the computational requirements typically increase superlinearly with the number of unknowns, and as such, all practical methods strive for keeping $h$ as large as possible in order to compute solutions quickly.

Due to the nonlinear nature of flow equations and the inherent truncation errors of numerical schemes, varying resolutions often lead to very significant differences in solutions. Such differences are not readily interchangeable across the different resolutions, as especially higher frequency content can be difficult to represent on a coarse grid. In order to improve the accuracy of a numerical method, we propose to learn a correction function that compensates for discretization errors on coarse grids and is representable by a NN. More specifically, we employ a CNN (Krizhevsky et al., 2012) as the correction should be translation invariant for generalization. We target time-dependent problems for which we infer a per time step correction. This is particularly challenging as approximation errors accumulate over time. In this work, we will focus on finite difference approximations (Strikwerda, 2004).

A discretized representation $\mathbf{s}$ of a smooth target function $f$ typically consists of a set of discrete physical quantities. Together, we call these discrete quantities $\mathbf{s}$ a state with which our correction approach can be formulated as follows:

$$\mathbf{s}^{n+1} = F(\mathbf{s}^n) + \mathbf{c}^n \tag{1}$$

where $F$ denotes a numerical solver for the target function $f$ moving a state $\mathbf{s}^n$ forward in time to a new state $\mathbf{s}^{n+1}$. Here, $\mathbf{c}^n$ denotes the correction function. Note that $\mathbf{c}^n$ potentially updates only a part of the full state, e.g., the velocity field $\mathbf{v}$.

We target solutions of the Navier-Stokes equations, which for incompressible Newtonian fluids takes the form

$$\frac{\partial \mathbf{v}}{\partial t} + \mathbf{v} \cdot \nabla \mathbf{v} = -\frac{1}{\rho}\nabla p + \nu \nabla \cdot \nabla \mathbf{v} + \mathbf{g} \quad \text{subject to} \quad \nabla \cdot \mathbf{v} = 0, \tag{2}$$

where $\rho$, $p$, $\nu$, and $g$ denote density, pressure, viscosity, and external forces, respectively. The constraint, $\nabla \cdot \mathbf{v} = 0$, is particularly important as it restricts motions to the space of divergence-free (i.e., volume preserving) velocities.

### 3.1 CORRECTION FUNCTION

The natural follow-up question is how to compute and represent the potentially highly nonlinear correction function of Eq. 1. Thanks to their flexibility and ability to represent nonlinearities, we use fully convolutional networks as our model for the correction function. A fine discretization, optionally with advanced numerical schemes, is employed to compute the reference data. Our goal is, then, to find an optimal approximation of the correction function for the given reference data. We will first approach this problem in a supervised manner with human intervention and then explain an improved approach that enables a NN model to learn the target function in an unsupervised manner.

As the velocity $\mathbf{v}$ is the key quantity for fluids, we target corrections $\mathbf{c}$ of this function in the following. However, our approach likewise could be applied to other functions. The correction is applied additively, i.e., $\mathbf{v} + \mathbf{c}$, and should be computed such that this sum matches the reference velocity $\mathbf{v}_R$, i.e., ideally $\mathbf{v} + \mathbf{c} = \mathbf{v}_R$. In practice, we will not aim for an equality but rather match the reference as closely as possible. As the reference is typically best defined in terms of a fine discretization with reduced approximation errors, it might have an altogether different resolution. In this case, care is needed when transferring functions between different discretizations especially when additional constraints such as physical laws of conservation need to be taken into account. We address this problem via an optimization and aim for minimizing the distance between representations on both grids with respect to a suitable metric.

For two different dimensionalities $\xi, \chi \in \mathbb{N}$ with $\xi < \chi$, consider two vector spaces $\mathbf{H} \in \mathbb{R}^\chi$ and $\mathbf{L} \in \mathbb{R}^\xi$ that both conserve volume (following Eq. 2), i.e., $\nabla \cdot \mathbf{h} = 0$ for $\forall \mathbf{h} \in \mathbf{H}$, and $\nabla \cdot \mathbf{l} = 0$ for $\forall \mathbf{l} \in \mathbf{L}$. Having a finer vector field $\mathbf{c}_H$, which contains the necessary information, we aim to find the closest vector field $\mathbf{c}_L$ ($\in \mathbf{L}$) to $\mathbf{c}_H$ ($\in \mathbf{H}$). We first describe how to transfer functions between both spaces in general before explaining how to find those that are particularly amenable for learning. Consider an interpolation operator $\mathbf{W}$ that introduces new data points within a vector field $\mathbf{c}_L$ ($\in \mathbf{L}$), i.e., $\mathbf{W}\mathbf{c}_L \in \mathbb{R}^\chi$. We, then, strive to minimize the distance between $\mathbf{W}\mathbf{c}_L$ and $\mathbf{c}_H$ such that $\mathbf{c}_L$ can best represent the information of $\mathbf{c}_H$ without loosing its volume conserving properties. Thus, we aim for computing $\mathbf{c}_L$ with

$$\underset{\mathbf{c}_L}{\text{argmin}} ||\mathbf{W}\mathbf{c}_L - \mathbf{c}_H||^2 \quad \text{subject to} \quad \nabla \cdot \mathbf{c}_L = 0. \tag{3}$$

This represents a constrained optimization problem with equality constraints, which we can solve via Lagrange multipliers $\lambda$ as follows:

$$\Phi = ||\mathbf{W}\mathbf{c}_L - \mathbf{c}_H||^2 + (\nabla \cdot \mathbf{c}_L)^\top \lambda. \tag{4}$$

This results in a system of equations as follows:

$$\begin{bmatrix} \mathbf{W}^\top \mathbf{W} & -\nabla \\ -\nabla^\top & 0 \end{bmatrix} \begin{bmatrix} \mathbf{c}_L \\ \lambda \end{bmatrix} = \begin{bmatrix} \mathbf{W}^\top \mathbf{c}_H \\ 0 \end{bmatrix}. \tag{5}$$

Using the Schur complement, we can simplify this system to speed up calculations:

$$\nabla^\top (\mathbf{W}^\top \mathbf{W})^{-1} \nabla \lambda = \nabla^\top (\mathbf{W}^\top \mathbf{W})^{-1} \mathbf{W}^\top \mathbf{c}_H, \tag{6}$$

$$\mathbf{c}_L = (\mathbf{W}^\top \mathbf{W})^{-1}(\mathbf{W}^\top \mathbf{c}_H - \nabla \lambda). \tag{7}$$

A NN, then, infers the correction $\hat{\mathbf{c}}_L$ by minimizing the supervised loss, $L_{\text{sup}} = \sum ||\hat{\mathbf{c}}_L - \mathbf{c}_L||$, for a pre-computed data set. While this method can efficiently yield divergence-free changes of

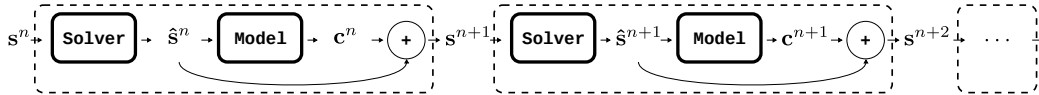

Figure 1: Overview of the recurrent training, in which the NN model is coupled with a differentiable PDE solver. Each unit, shown with dashed-lines, represents one time step, and multiple steps are unrolled so that the model can take into account temporal information when learning.

resolution, i.e., move a function such as the aforementioned correction from fine to coarse grids, the solutions obtained in this way are not necessarily optimal for deep learning methods as we will evaluate in more detail below. Hence, we first introduce an extension that takes into account the temporal evolution of the correction function.

### 3.2 TEMPORAL REGULARIZATION OF CORRECTIONS

The vector fields we target are obtained from a numerical simulation. Here, the underlying flow equations are solved for a finite number of steps from an initial condition. The simulation produces a series of states, which represent the spatio-temporal changes of fluid volume and velocity. Given the setup as described so far, we can acquire the correction of a basic numerical simulation by running the same simulation with a reduced step size in space and time to reduce approximation errors. In addition, we can employ more accurate numerical schemes for the reference as we will demonstrate below. Having both the basic simulation velocity field $\mathbf{v}_B$ and the reference simulation velocity field $\mathbf{v}_R$, we can compute the correction vector field $\mathbf{c}_L$ via Eq. 7 for $\mathbf{c}_H = \mathbf{v}_R - \mathbf{W}\mathbf{v}_B$.

As our aim is to find a learned representation of the corrections, a crucial aspect to consider is the sensitivity (Murphy et al., 2004) of the targeted function (i.e., the correction) with respect to the data at hand, i.e., in our case, the state of a coarse simulation. Here, we observe that the correction vector fields vary strongly in time even for smooth changes of the basic simulation. That means the correction function has a very nonlinear and difficult to learn relationship with the observable data in a simulation.

In order to address this difficulty, here, we introduce supervision in data acquisition by employing a temporal regularization that makes the correction function learnable without sacrificing its correcting properties. As we are dealing with continuous models, we know that the flow motion changes smoothly in time if it is resolved finely enough. As our solution changes in space as well as time, a learned function has to be able to represent spatial as well as temporal changes of the target function correctly. We focus on the inference of per-time step corrections, and thus it is especially important to obtain correction functions that change smoothly over time (spatial regularization could potentially also be incorporated during learning implicitly or explicitly). Consequently, we regularize our correction vector fields such that they change smoothly in time by penalizing temporal change of the correction vector field. These are given by $d\mathbf{c}_L/dt$, which we minimize together with the transfer from fine to coarse discretizations:

$$\operatorname*{argmin}_{\mathbf{c}_L} \left( ||\mathbf{W}\mathbf{c}_L - \mathbf{c}_H||^2 + \beta||\frac{d\mathbf{c}_L}{dt}||^2 \right) \quad \text{subject to} \quad \nabla \cdot \mathbf{c}_L = 0. \tag{8}$$

Here, $\beta$ is the regularization coefficient. This yields a new system of equations as follows:

$$\begin{bmatrix} \mathbf{W}^\top\mathbf{W} + \beta\frac{2}{\Delta t}\mathbf{I} & -\nabla \\ -\nabla^\top & 0 \end{bmatrix} \begin{bmatrix} \mathbf{c}_L \\ \lambda \end{bmatrix} = \begin{bmatrix} \mathbf{W}^\top\mathbf{c}_H + \beta\frac{2}{\Delta t}\mathbf{c}_L^{n-1} \\ 0 \end{bmatrix}, \tag{9}$$

where $\Delta t$ is the timestep size, $\mathbf{I}$ is the identity matrix, and the superscript $n - 1$ of $\mathbf{c}_L$ denotes the state at the previous step in time. We used the $\beta = 0$ in the following experiments and found it effective for both the correction accuracy and training.

### 3.3 UNSUPERVISED LEARNING WITH DIFFERENTIABLE PHYSICS

So far, we have described how to acquire a learnable data set obtained with human support for introducing temporal regularization. The NN can, then, learn the corrections given the low-resolution input simulations, i.e., minimizes the error of inferred corrections. This naturally requires special

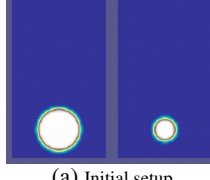 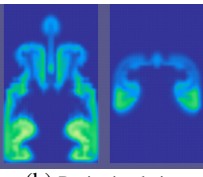 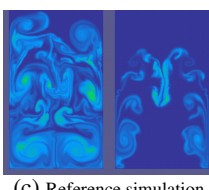 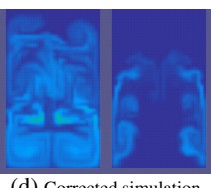

(a) Initial setup      (b) Basic simulation      (c) Reference simulation      (d) Corrected simulation

Figure 2: Rising smoke. (a) The initial size of a smoke volume is randomized in two simulations. After 1,000 steps, the same initial setup results in significant differences between (b) the basic simulation and (c) the reference simulation. Moreover, the small variations in the initial volume result in very different density configurations. (d) The basic version is assisted by the ground truth correction from Eq. 9 without NN inference.

care in terms of choosing right amount of temporal regularization, which in practice is strongly problem dependent. To reduce the reliance on human intervention, we propose a training approach with which the model can directly learn the target function by recurrently observing the temporal dynamics of the physical model as illustrated in Fig. 1. The recurrent training can be achieved via differentiable physics solvers that allow for a back-propagation of gradients through the discretized physical model. In this way, we can let a NN learn the sought-after correction function in an unsupervised manner. As we will demonstrate below, the main advantage of this approach is that the NN experiences how corrections influence the evolution of the dynamics and receive gradients in order to improve inference accuracy. In the following, we employ a differentiable Navier-Stokes solver from concurrent work (anonymous, 2020). This solver builds on the automatic differentiation of a machine learning framework to compute analytic derivatives and augments them with custom derivatives where built-in derivatives would be inefficient. This setup allows for straightforward integration of solver functionality with machine learning models and enables end-to-end training in recurrent settings.

Denoting the discretized physical model as $F_m$, we can define our loss as $L_{\text{un}} = \sum_i^m ||(F_m(\mathbf{s}^{t_0+i}) + \mathbf{c}_L) - \mathbf{M}\mathbf{s}_H^{t_0+i}||^2$, where the notation of the NN inference for $\mathbf{c}_L$ is omitted for brevity, $\mathbf{M}$ is a sampling operator to make the spatial discretizations consistent, and $m$ denotes the number of recurrent steps. Above, each loss evaluation starts at time $t_0$. Note that a state $\mathbf{s}^{t_0+i}$ depends on all $i-1$ previous states, and thus requires a back-propagation of the gradients through $i-1$ solver evaluations. In contrast to the supervised approach, we can directly evaluate the difference between the corrected simulation result, as provided by the unsupervised loss formulation, and the reference state $\mathbf{s}_H$. In practice, this can be achieved without explicitly preparing for learnable input-output pairs, which are acquired using our optimization for training. Once sequences of reference simulations are collected, we can directly use the sequences by selecting an initial frame $t_0$, which is downsampled to the basic solver's resolution and fed to the architecture as an initial input. During each recurrent step, the intermediate results are evaluated via the differentiable solver and the current state of the correction model. These intermediate results are compared with the given consecutive reference frames, and the model can update its weights accordingly. With this setup, the model can directly learn from simulation results that have experienced inferred corrections via the unrolled recurrent steps, each of which incorporates the physics solver.

## 4 EXPERIMENTS

To acquire our data sets, we generate a set of simulation sequences, which are started from randomized initial setups according to the given parameters. These sequences are used for obtaining pairs of input and correction velocity fields to training our NN. Anonymized and time-stamped supplemental material for our submission can be downloaded at `https://www.dropbox.com/sh/bl2xmrtc2loot14/AADChTpswLYmy7Yq_PG1zfXka?dl=0`.

### 4.1 SUPERVISED CORRECTION OF RISING SMOKE

This example encompasses a volume of hot smoke rising from the bottom of an enclosed container. The motion of the smoke volume is driven by buoyancy forces computed from the density field

| Basic model | Ground truth | Our model | Basic model | Ground truth | Our model |

(a) Test A                                    (b) Test B

Figure 3: Corrected rising smoke simulation. Our model is applied to two test cases A and B. After correcting 400 steps, the density fields are compared among the basic method, ground truth, and corrected method (our model). (See Fig. 11 in Appx. A.2 for more comparisons.)

using the Boussinesq model. We randomize the initial size of the smoke volume. Fig. 2 shows two selected randomized initial setups at the left most. Additionally, two different results of the reference and basic simulations after 1,000 simulation steps are shown in the subsequent columns, respectively. It is worth noting that the reference simulations are significantly different from the basic ones, moreover, the small variations in the initial setup result in significant differences even after short periods of time. We use a regular MacCormack scheme (Selle et al., 2008) for the basic simulation and the more accurate and compute-intensive advection-reflection scheme (Zehnder et al., 2018) for the reference. The grid resolution of the basic simulation is $32 \times 64$, and a four times finer grid is used for the reference.

In the following, we denote the basic version corrected by the full correction function computed with Eq. 9 as the "ground truth" version. The coarse version naturally cannot exactly represent the high-resolution reference, and as such, we consider the corrected version without NN inference as the "ground truth" version we are aiming for with our model. The ground truth of our selected two simulations is shown at the right most in Fig. 2.

We train our model using the data collected from the 20 simulations; each simulation performs 1,000 steps, thus we collected 20,000 samples in total. From each simulation, we extract pairs of input features $\mathbf{x}_L$ and a correction vector field $\mathbf{v}_L$. For the input features, we can use multiple channels depending on the complexity of our target function. From a pilot test, we experimentally found that nine channels, i.e., three subsequent states give best results for our problem (see Fig. 10 in Appx. A.2). These consist of the velocity and density fields, i.e., $\mathbf{x}_L = \{\mathbf{v}_L^{n-2}, \mathbf{v}_L^{n-1}, \mathbf{v}_L^n, \rho_L^{n-2}, \rho_L^{n-1}, \rho_L^n\}$. We randomly split the data into a training data set of 95% and a validation data set of 5%. We use network model A (see Appx. A.1 and Fig. 9a) and train for 200 epochs with a batch size of 32. We use the $L_{\text{sup}}$ loss and an Adam optimizer (Kingma & Ba, 2014) with an adaptive learning rate starting at 0.001.

We test the trained model within two simulations that are not part of the training data set. A selected frame for these tests is shown in Fig. 3, which visualizes the density fields. Our model produces significantly closer results to the ground truth than the basic simulation. Since the error accumulates over time, differences become more apparent in later frames. Nevertheless, we find that, despite the large number of nonlinear simulation steps, the inferred correction field manages to move the solution close to the desired state. The average errors in both density and velocity fields are shown in Fig. 4. Here, we show relative errors with respect to ground truth correction. The errors show that our supervised model corrects the simulations

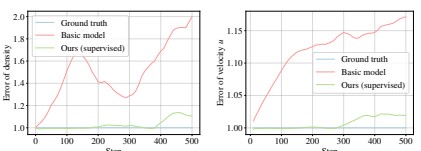

(a) Density        (b) Velocity $u$

Figure 4: Relative errors of the rising smoke correction model. The trained model corrects the velocity field for 500 steps. Relative errors in (a) density and (b) velocity with respect to reference are shown for Test A from Fig. 3.

very well; errors stay very close to the ground truth values and start to slightly deviate only after 400 steps of simulation.

## 4.2 NAIVE CORRECTION

We now evaluate the effect of the proposed optimization approach to compute the correction function on the learning process. In our context, a straightforward, i.e., "naive", way to get a correction is to directly downsample the reference correction (i.e., $\mathbf{c}_H$) to the target domain. We find that this

correction function produces comparable ground truth results for the rising smoke, and hence, we can train a network in a supervised manner with this downsampled correction data. Fig. 5a shows an example. Here, we directly apply a regularized and a "naive" correction function without involving the learning step. This figure highlights that both versions yield almost identical results and, hence, the regularization also retains the important characteristics of the high-resolution reference. Apart from omitting our optimization step, i.e., providing an unregularized set of target data for the supervised training, all steps are performed as described above.

Figs. 5b–5d show the results of training with both versions. The "naive" NN model converges to a much higher overall level of error, and the resulting function approximation is unusable in practice. The inference errors quickly cause the basic simulation to deteriorate and become un-

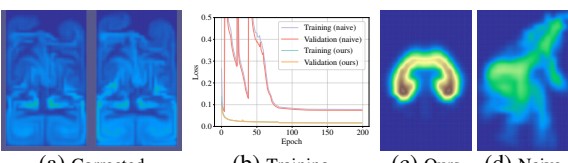

| (a) Corrected | (b) Training | (c) Ours | (d) Naive |

Figure 5: Comparison between the naive and our model.

stable. This behavior contrasts with to our approach where the temporal regularization makes it possible to accurately represent the correction function with the CNN. We also demonstrate that our approach generalizes to other problems in the context of fluid flows. See Appx. A.3 for a liquid stream example.

We have shown that the smooth changes of a correction field with respect to smoothly changing inputs are crucial for successfully learning the correction function. Despite obtaining gains in accuracy from the correction function, it requires manual input to make the correction function amenable to learning via suitable temporal regularization. In the following, we will show how our unsupervised learning approach alleviates this difficulty.

## 4.3 LEARNING WITH A DIFFERENTIABLE PDE SOLVER

We start with a test for the reference data that presents relatively smooth motions close to the basic simulation. Hence, we first test the correction from a low-resolution to a high-resolution simulations with the same standard numerical scheme for both versions. In this test, we use a semi-Lagrangian advection (Stam, 1999), and the reference data is generated for 100 steps from 20 randomized initial sizes of the smoke volume, i.e., 2,000 samples in total. It is worth noting that the actual amount of samples the NN model sees is much larger, because the intermediate output states from the recurrent iterations are likewise seen by the model. These states vary over the course of the training steps since the applied correction function changes.

Fig. 6 shows the evaluation of our trained model. It is apparent that our model with 48 recurrent steps yields a model that successfully removes the majority of the errors as shown on the left of Fig. 6. The graphs of Fig. 6a show velocity errors relative to the ground truth version for each of three models trained with different numbers of recurrent steps. Here, we can see that, with more recurrent steps, the model has a higher chance to learn the correction function for long term accuracy. Fig. 6b shows errors of inferences for the given 100 ground truth steps, where the ground truth correction is evaluated via the difference between the solver result and the downsampled reference data. These graphs demonstrate that the model with 48 recurrent steps yields the best long term accuracy despite

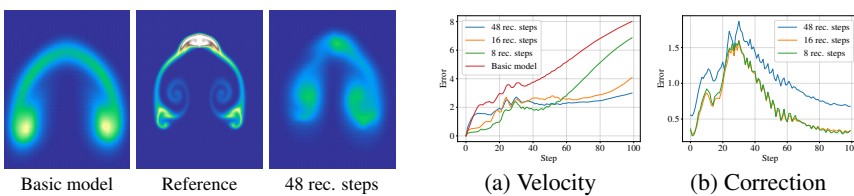

| Basic model | Reference | 48 rec. steps | (a) Velocity | (b) Correction |

Figure 6: Analysis of the corrected simulations for rising smoke. Example frames after 80 timesteps for the following versions are shown: basic, reference, and our model. Our model is trained with 48 recurrent steps as in Fig. 1. The trained model, then, corrects the velocity field for 100 steps. The $L^2$ errors in (a) velocity and (b) correction are measured from the ground truth at every step. (See Fig. 12 in Appx. A.2 for more results.)

containing larger per step differences with respect to ground truth corrections. This indicates that, by looking ahead more steps from the solver, the model learns to anticipate the future dynamics and corrects the target function accordingly instead of relying on the seemingly ideal corrections for each time step.

Next, we test our model in a more complex settings using the reference data used in Sec. 4.1. Fig. 8 shows the corrected results of our unsupervised model, which was again trained with 48 recurrent steps, and relative errors are given in Fig. 7. It is worth noting that, within the first ca. 300 steps, the unsupervised model learns even more accurate corrections than the target ground truth version. This illustrates that the model can learn from the gradients provided over the course of the many timesteps by the differentiable solver to find a correction function that performs better than the carefully precomputed supervised version. On the other hand, we also see that the gains

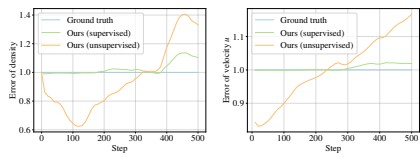

(a) Density      (b) Velocity $u$

Figure 7: Relative errors of both supervised and unsupervised models. Relative errors in (a) density and (b) velocity with respect to reference are shown for Test A from Fig. 8.

disappear over the course of very long sequences (more than ca. 500 steps). Presumably, the unsupervised model would need to be trained with more recurrent steps to learn to anticipate very long-term changes.

## 5 DISCUSSION AND CONCLUSION

A key benefit of our approach is the gain in performance resulting from our trained models. A correction velocity field for a given input is inferred only on the basic simulation grid, i.e., the low-resolution grid. This inference happens for each solving step to assist the underlying numerical solver and only requires a fixed $\mathcal{O}(n)$ cost for $n$ degrees of freedom. For example, for our rising smoke test, a simulation involving the trained NN model took ca. 20 seconds for 1,000 steps whereas its corresponding high-resolution counterpart took ca. 104 seconds to compute. See Table 1 in Appx. A.4 for more details.

At training time, the unsupervised learning approach leads to significantly longer training times since each recurrent step of the architecture can require evaluating a complex numerical procedure. A potential remedy and interesting topic for future work would be to use larger timestep size in the solver such that the model could directly learn the correction for longer horizons. As the excellent performance of the unsupervised model for the initial stages of our simulations suggests, this is a very promising avenue.

To summarize, we introduced a novel approach that assists numerical methods by learning a correction function to improve the accuracy of the solution. We demonstrated that taking into account the temporal information is crucial for our goal and an optimization step can ensure that sequences of corrections can be learned accurately by a NN model. The model successfully improves the accuracy for previously unseen PDE solves. Additionally, an unsupervised training via a differentiable solver can be employed to further improve the learned correction for time spans that do not strongly exceed the number of steps seen during training. Despite focusing on fluids, we envision that our approach can be applied to a variety of other application domains that involve numerical methods for spatio-temporal problems, from plasma physics Lewis & Miller (1984) to climate modeling Randall et al. (2007).

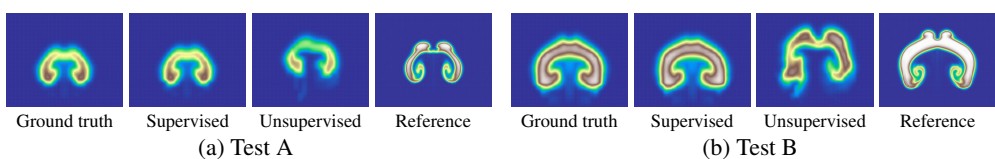

Ground truth   Supervised   Unsupervised   Reference     Ground truth   Supervised   Unsupervised   Reference

(a) Test A                       (b) Test B

Figure 8: Corrected rising smoke simulations. Both supervised and unsupervised models are applied to two test cases A and B. After correcting 200 steps, the corrected density fields are compared with respect to the ground truth and the reference.

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

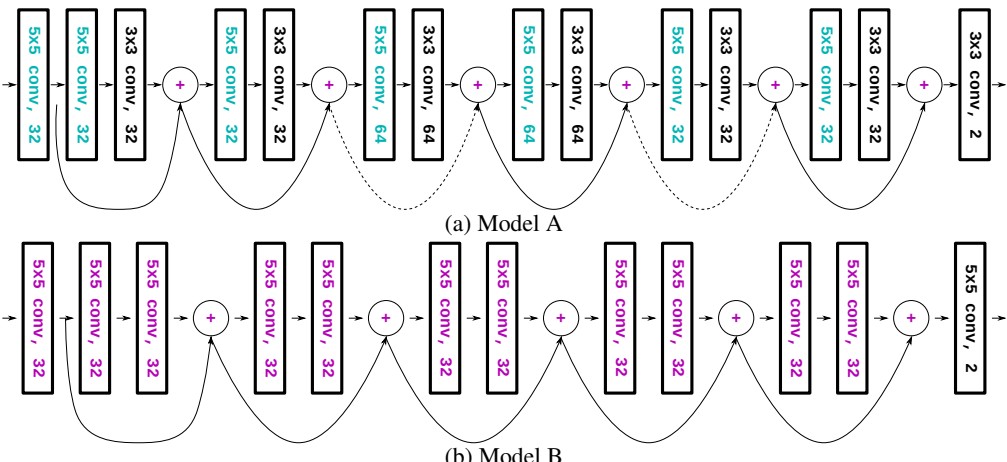

(a) Model A

(b) Model B

Figure 9: NN architectures. Two models that are used in our experiments are presented. The blue and magenta colors indicate the ReLU and LeakyReLU activation functions, respectively. The curved lines indicate skip connections; the dotted lines show an additional $1{\times}1$ convolution, which makes the number of channels same when adding.

## A    APPENDIX

### A.1    NEURAL NETWORK MODEL

For the NN architecture to represent the correction function $\mathbf{c}^n$ of Eq. 1, we use a fully convolutional network with residual blocks (He et al., 2015). The two network models, which are used for our experiments, are shown in Fig. 9. The models A and B consist of 405K and 265K training parameters, respectively. Hence, we use the model B for setups with a smaller amount of training data. Our models are implemented using the *TensorFlow* framework (Abadi et al., 2015).

### A.2    RISING SMOKE EXPERIMENTS

In order to decide an effective input feature for a NN model, we compare two models trained with different numbers of input channel. Fig. 10 shows the errors of inferred correction between two models and indicates that the model trained with three subsequent states (i.e., $k = 3$) performs slightly better particularly for longer steps. Nevertheless, we observe that the difference is negligible.

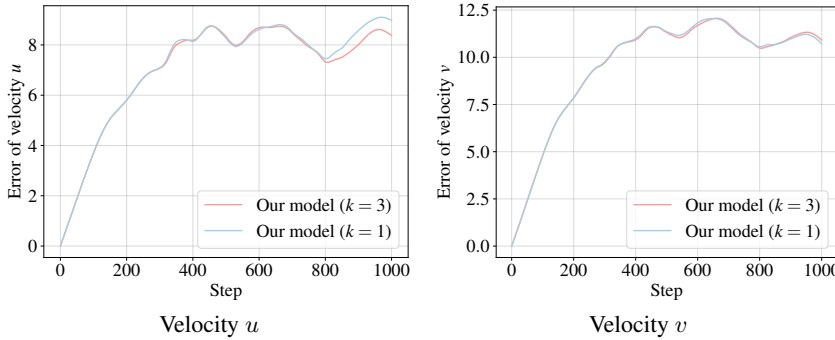

Velocity $u$

Velocity $v$

Figure 10: Comparison of errors in inferred correction between two supervised models with different numbers of input channels. Here, $k$ denotes the number of subsequent states used for the input.

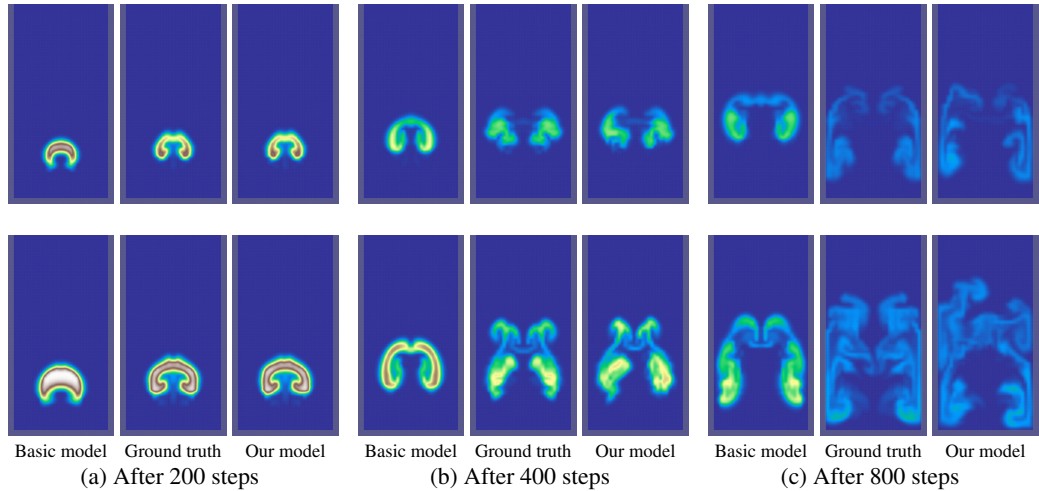

(a) After 200 steps      (b) After 400 steps      (c) After 800 steps

Figure 11: Corrected rising smoke simulation. Our supervised model is applied to two test cases shown at each row. At three selected steps, the density fields are visualized among the basic method, ground truth, and corrected method (our model).

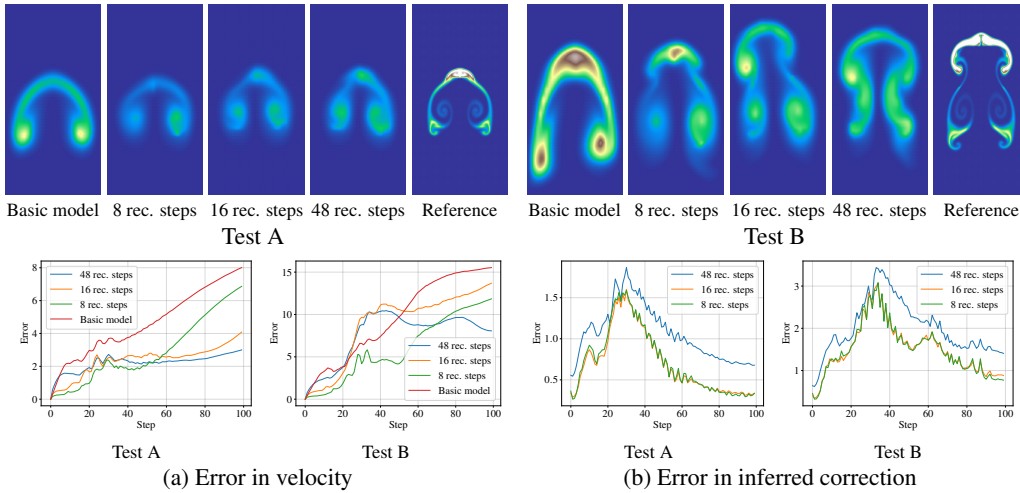

(a) Error in velocity      (b) Error in inferred correction

Figure 12: Two test simulations for the rising smoke example. After individual training with different numbers (i.e., 8, 16, and 48) of recurrent steps, each model is applied to two test setups, test A and B. The images at the top row show the density visualization after 80 steps. The graphs in (a) compare the corrected velocity fields, and those in (b) show the errors of inferences for the given input steps.

## A.3 LIQUID STREAM WITH STEP OBSTACLES

As an extension of our method to other types of fluid simulations, we target a liquid stream flowing over a backward facing step. To make the setup slightly more complex, we introduce a second step on the right side of the domain. Height and width of both steps are randomized to generate data sets. We note that Lagrangian approaches, i.e., particle-based discretizations, form an important class of fluid-related problems and our approach also generalizes to such approaches via an auxiliary grid. This example demonstrates how to use our method in conjunction with the methods such as the *fluid implicit particle* algorithm (Brackbill et al., 1988). In this method, the flow motion is transferred to a helper grid in every simulation step and follow the remaining steps as explained before. Fig. 13 shows this setup for two different initial conditions. Here, the color indicates the direction of velocity such that it highlights the locations of vortex structures in the flow. All simulations adopt the APIC method without surface tension but instead with a gravitational force and a fixed inflow velocity.

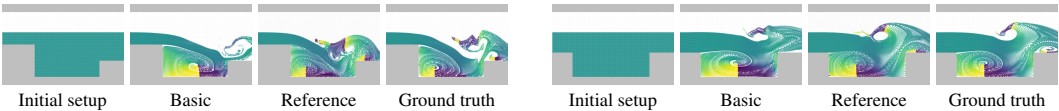

Initial setup      Basic      Reference      Ground truth     Initial setup      Basic      Reference      Ground truth

Figure 13: Stream flow starting from a randomly initialized setup for the width and height of two obstacles. The color indicates the direction of velocity, in each case, for basic, reference, and ground truth versions.

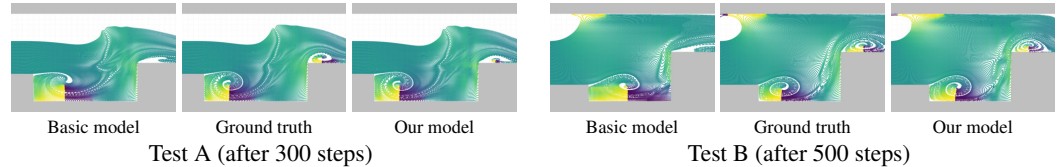

Basic model      Ground truth      Our model     Basic model      Ground truth      Our model

Test A (after 300 steps)        Test B (after 500 steps)

Figure 14: Corrected stream flow simulations. Our model is applied to two test cases A and B. The density fields are compared at the selected steps. The color indicates the direction of velocity.

The grid resolution for the basic and corrected simulations is $32 \times 16$, and the reference grid is four times finer.

Comparing the reference and basic simulations, we can find that the basic simulation has difficulties resolving the rotating motions that form behind the step geometries. We use the network model B (see Appx. A.1 and Fig. 9b) and train the model with the data from ten simulations. Fig. 14 shows the results of our model applied to two test cases. A notable improvement of our model is that it manages to reintroduce the vortex above the right step that was lost in the basic version. The shape of the left vortex also improves to match the ground truth version.

## A.4 PERFORMANCE

Here, we give details about the performance of our method to illustrate the gains that can be achieved by evaluating our trained model. Table 1 shows the timings measured from different simulations of each example. All experiments were performed on an Intel Xeon E5-1620 3.70 GHz processor with 128 GB memory. The trained NN models were evaluated with the CUDA support and TensorFlow on an NVIDIA GeForce GTX 960 GPU with 4 GB video memory. The reference is a full simulation with four times higher resolution evaluated with an optimized CPU-based solver (the same one using to generate the input for our method).

Table 1: Performance comparison. Simulation timings (in seconds) are measured for 1,000 simulation steps. Here, *un.* and *sup.* denote the unsupervised and supervised learning models, respectively.

| Example | Test | Basic | Reference | Ours | |
|---|---|---|---|---|---|
| Rising smoke (Fig. 8) | A | 9.40 | 104.46 | 17.56 (un.) | 19.91 (sup.) |
| | B | 9.87 | 117.21 | 17.12 (un.) | 18.96 (sup.) |
| Liquid stream (Fig. 14) | A | 18.00 | 85.49 | | 29.86 (sup.) |
| | B | 19.58 | 84.10 | | 31.82 (sup.) |

