# OpenReview forum: "Learning Time-Aware Assistance Functions for Numerical Fluid Solvers"
_ICLR.cc/2020/Conference — Reject_

### Official Review · AnonReviewer2 · 2019-10-18
**Official Blind Review #2**

**Rating:** 6

**Review:**

The authors aim at improving the accuracy of numerical solvers (e.g. for simulations of partial differential equations) by training a neural network on simulated reference data. The neural network is used to correct the numerical solver. For different tasks they set up an approximation scheme via minimizing a square loss plus a task specific regularization (e.g. volume preservation in the Navier-Stokes equation example). This is then trained in a supervised manner. They also explore an unsupervised version by back-progagating through a differentiable numberical solver.
The proposed method seems straight forward, but effective as the simulations seem to indicate.

**Experience Assessment:**

I do not know much about this area.

**Review Assessment: Checking Correctness Of Derivations And Theory:**

I assessed the sensibility of the derivations and theory.

**Review Assessment: Checking Correctness Of Experiments:**

I assessed the sensibility of the experiments.

**Review Assessment: Thoroughness In Paper Reading:**

I read the paper at least twice and used my best judgement in assessing the paper.

---

> ### Author Response · Authors · 2019-11-12
> **Response to Review 2**
>
> Thank you very much for your time to review our paper and your positive assessment.

---

### Official Review · AnonReviewer1 · 2019-10-23
**Official Blind Review #1**

**Rating:** 3

**Review:**

The paper proposes learning NN to correct for inaccuracies in numerical solvers of PDEs, with experimental focus on fluid flow simulation. It lists two approaches: (1) compute correction in high resolution simulation from reference states, convert to low-resolution correction, and train NN to predict low-res correction (optionally with temporal regularization, and (2) directly simulate forward using correction prediction and differentiable PDE solver and optimize to match the given reference states. It shows empirical results on better approximating fluid flow simulation.

The paper tackles an important problem of using NN to speed up expensive simulation computations. The main limitation appears to be the significance of machine learning approaches. Approach (1) is a naive prediction using NN, and approach (2) involves interesting differentiation through PDEs, but that’s directly borrowed from another concurrent anonymized submission. This paper alone does not seem to have enough novel ML contributions for acceptance.

The writing can benefit from more clarity and better structure. For example,
- Write prediction as \hat{c}_L(s) to show what is input for the NN
- Put definition for c_H in Section 3.1. It seems to be based on v_R and v_B but would benefit from being explicit.
- If beta=0 is used, why do you need Section 3.2? Is it a typo?
- In Section 3.3, I am unsure if it should be called unsupervised, as you are given reference state s_H. How is the assumption different from Sections 3.1 and 3.2? Also, a relevant reference [1]

[1] Bengio, Samy, et al. "Scheduled sampling for sequence prediction with recurrent neural networks." Advances in Neural Information Processing Systems. 2015.



**Experience Assessment:**

I do not know much about this area.

**Review Assessment: Checking Correctness Of Derivations And Theory:**

I assessed the sensibility of the derivations and theory.

**Review Assessment: Checking Correctness Of Experiments:**

I assessed the sensibility of the experiments.

**Review Assessment: Thoroughness In Paper Reading:**

I read the paper at least twice and used my best judgement in assessing the paper.

---

> ### Author Response · Authors · 2019-11-12
> **Response to Review 1**
>
> Thank you very much for taking the time to review our paper. We would like to clarify a set of points below.
>
> Our second approach, which we call unsupervised, could be adapted to a variety of PDE problems. The key advantage of this approach is that it directly integrates an NN model for the proposed correction with a numerical solver to be improved. This also means that, in fact, you do not need to provide the reference state s_H since we can also integrate an advanced solver with a higher resolution configuration and acquire the reference data on the fly. Nevertheless, it is natural to precompute the reference data since it is very time consuming. We reused the reference data for our target problem; however, this is not a prerequisite.
>
> Yes, beta=0 is a typo; it should be beta=1 instead. Thank you for the suggestions. We would be happy to incorporate your suggestions for better exposition.

---

### Official Review · AnonReviewer3 · 2019-10-26
**Official Blind Review #3**

**Rating:** 3

**Review:**

Numerical solvers for partial differential equations take a lot of time to get high resolution results since they have to explore high dimensional grid in function domain. Thus, it is important to interpolate between grids to get high resolution results. In this paper, the authors propose the model that assists PDE solver by correcting residuals in a data-driven way. Specifically, they try to approximate NN to correction function in supervised and unsupervised manners. They also propose a temporal regularization method that smooths behavior of fluid between times. As a result, proposed method can generate high resolution results in efficient way with smoke rising simulation dataset.

Significance
- The field that the paper target is too narrow so that it may not be significant to general machine learning community.

Novelty
- I think this approach is somewhat novel since it introduces new direction in the field where the model assists PDE solver to improve performance with learned correction function.

Clarity
- I think this paper is well written in most places, but the paper is not intended to make it easy for general machine learning researchers to understand and follow the problem thoroughly.

Pros
- The proposed method is orthogonal to other methods such as super-resolution and may be complemented to other existing approaches.

Cons
My main concern is that the evaluation of proposed method is quite limited in the sense that- The method is evaluated with only single dataset.- Only basic model is used to compare to proposed model. What about other baseline models use similar deep learning approaches?
- There should be various PDE problems other than Navier-Stokes to evaluate the general effectiveness of the proposed model.
- I think time comparison/analysis should be provided since the paper targets to get efficient approximation (using NN) in numerical analysis field.

Questions and comments
- I wonder the affect of hyper-parameter in temporal regularization.
- I wonder performance comparison to existing NN based approaches.

**Experience Assessment:**

I do not know much about this area.

**Review Assessment: Checking Correctness Of Derivations And Theory:**

I did not assess the derivations or theory.

**Review Assessment: Checking Correctness Of Experiments:**

I did not assess the experiments.

**Review Assessment: Thoroughness In Paper Reading:**

I made a quick assessment of this paper.

---

> ### Author Response · Authors · 2019-11-12
> **Response to Review 3**
>
> Thank you very much for your time to review our paper. We would like to address your concerns in the following.
>
> Although we demonstrated our approach with a particular example, i.e., the Navier-Stokes equations in the paper, our correction approach is not limited to this domain. A variety of PDE problems and their numerical solvers share the same goal for better accuracy and better efficiency, and the proposed approach can be generally applicable to those problems. We demonstrated our correction approach within two very different types of fluid flows, i.e., smoke and liquid. The fluid flows that we target with our work are a particularly complex and relevant problem.
>
> As you point out, a set of NN-based methods were proposed to achieve enhanced results for similar problems; one popular approach is super-resolution. Such a super-resolution approach synthesizes the fine details based on the low resolution input, which is left unmodified and potentially inaccurate due to the limited resolution. On the other hand, our approach directly assists (i.e., corrects) the given numerical solver itself such that its solving accuracy improves with respect to the reference solver’s solution. As we show in our paper, the evolution of smoke in the reference simulation significantly differs from its basic version. Our approach is orthogonal to such a posteriori methods like super-resolution algorithms, and thus both could be used together to yield further improvements.
>
> The hyperparameter beta in our supervised approach tries to balance between the quality of correction and the difficulty of learning task by an NN model. In our experiments, the NN model failed to learn the correction dataset for beta=0. On the other hand, for beta=10, the quality of ground truth correction deteriorated, which is why we empirically chose beta=1. We would be happy to add these experiments and the discussion to our paper.

---

### Decision · Program_Chairs · 2019-12-19

**Decision:**

Reject

**Comment:**

This paper provides a data-driven approach that learns to improve the accuracy of numerical solvers. It solves an important problem and provides some promising direction. However, the presented paper is not novel in terms of ML methodology. The presentation can be significantly improved for ML audience (e.g., it would be preferred to explicitly state the problem setting in the beginning of Section 3).